# Does Chlorination Promote Antimicrobial Resistance in Waterborne Pathogens? Mechanistic Insight into Co-Resistance and Its Implication for Public Health

**DOI:** 10.3390/antibiotics11050564

**Published:** 2022-04-22

**Authors:** Martins A. Adefisoye, Ademola O. Olaniran

**Affiliations:** Discipline of Microbiology, School of Life Sciences, College of Agriculture, Engineering and Science, Westville Campus, University of KwaZulu-Natal, Private Bag X54001, Durban 4000, South Africa; olanirana@ukzn.ac.za

**Keywords:** waterborne pathogens, antimicrobial resistance (AMR), chlorination, mutant selection window (MSW), resistance mechanisms, public health

## Abstract

Chemical agents including chlorine and antibiotics are used extensively to control infectious microorganisms. While antibiotics are mainly used to treat bacterial infections, chlorine is widely used for microbial inactivation in the post-secondary disinfection steps of water treatment. The extensive use of these agents has been acknowledged as a driving force for the expansion of antimicrobial resistance (AMR) and has prompted discourse on their roles in the evolution and proliferation of resistant pathogens in the aquatic milieus. We live in a possible “post-antibiotic” era when resistant microbes spread at startling levels with dire predictions relating to a potential lack of effective therapeutic antibacterial drugs. There have been reports of enhancement of resistance among some waterborne pathogens due to chlorination. In this context, it is pertinent to investigate the various factors and mechanisms underlying the emergence and spread of resistance and the possible association between chlorination and AMR. We, therefore, reflect on the specifics of bacterial resistance development, the mechanisms of intrinsic and acquired resistance with emphasis on their environmental and public health implications, the co-selection for antibiotic resistance due to chlorination, biofilm microbiology, and multidrug efflux activity. In-depth knowledge of the molecular basis of resistance development in bacteria will significantly contribute to the more rational utilization of these biocidal agents and aid in filling identified knowledge gap toward curbing resistance expansion.

## 1. Introduction

Antimicrobial-resistant waterborne pathogens are a major global health concern. There are rising reports of resistance selection in microorganisms due to inappropriate use of antimicrobial agents [1,2]. Upon antibiotic usage, a small portion of the microbial cells, even <10^−9^ within a susceptible wild-type population, is not affected by the antimicrobial agent. This subgroup retains antimicrobial impeding mutations, which favor their selection during treatment. Thus, antibiotic resistance is known to emerge during the frame of a selective compartment described as a mutant selection window (MSW) [3].

The MSW hypothesis was initially proposed based on a study investigating resistance to quinolones in Gram-negative bacteria in the 1990s [4]. The study noted the existence of a “dangerous” concentration range where mutant selection occurs most frequently and suggested that resistance prevention against norfloxacin in strains carrying a “first mutation” should employ older quinolones and emphasizes the need for early detection of such strains [5]. Accordingly, the MSW postulates that, for every antimicrobial drug-microorganisms combination, there exists a drug concentration range where resistant-mutants selective amplification occurs in a single step, leading to reduced susceptibility [6]. This range extends from the minimum inhibitory concentration of the drug, which prevents colony growth by 99% (MIC_99_), to the mutant prevention concentration (MPC), at which the least-susceptible single-step mutant subpopulation is inhibited. Within these limits, selective enrichment of resistant mutants frequently occurs [7]. Upon mutation acquisition, the MIC_99_ and the MPC increase, shifting up the MSW of an antimicrobial and making it more challenging to prevent the emergence of new mutant strains [8]. 

Some reports have documented correlations between antibiotics and chlorine resistance from aquatic environments [9,10], while other studies have reported the co-selection for antibiotic resistance among bacteria due to chlorination [11,12]. Although chlorination has proved efficient in reducing waterborne diseases, little effort has been devoted to exploring the potential impacts of chlorine disinfection in enhancing antibiotic resistance genes (ARGs). The apprehension about the contribution of chlorination to an upsurge of AMR in waterborne pathogens presents a potential threat to human health and the aquatic ecosystem. Albeit, the reasons for this phenomenon and the factors triggering AMR expansion during chlorination remain poorly understood [13].

To this end, this review describes the implications of intrinsic and acquired bacterial resistance with emphasis on aquatic and human health, connections between antibiotics and chlorine resistance, the roles of microbial biofilms and multidrug efflux activity in AMR, and identifies knowledge gap in the existing literature on the molecular biology of AMR. A comprehensive understanding of the molecular basis of resistance development in bacteria will significantly contribute to these agents’ more rational selection and utilization toward curbing AMR proliferation and advancing the global antimicrobial stewardship effort.

## 2. Water Quality and Human Health

Easy access to improved water sources is vital to human health and development. Contaminated water predisposes individuals to hazards and has been linked to disease outbreaks. Despite progress in global access to improved water quality in recent years, over two billion people still lack access to safely managed water sources [14]. Climate change, rising human population and urbanization pose challenge for water supplies and important strategies such as wastewater recycling is becoming vital in water reclamation. Inadequately treated recycled wastewater poses an unacceptable risk of infectious disease and a threat to public health [15]. 

Concerns about the microbiological quality of water have previously focused on the occurrence of pathogens. However, the prevalence of antimicrobial-resistant bacteria (ARB) and resistance genes in water sources is an emerging issue of public health globally [16]. Different studies have presented varying opinions on the selection or enrichment of resistant strains during chlorination [10,13]. Jin et al. [13] reported that chlorination promotes the emergence and exchange of resistance genes across bacterial genera. Murray [17] observed a significant increase in the level of multiple drug-resistant (MDR) bacteria in sewage, after treatment with chlorine, and during a laboratory-scale chlorination investigation. Contrarily, other studies have suggested that disinfection does not contribute to resistant strain selection but rather induces the development of antimicrobial resistance [18,19]. Another study yet has noted that the selection of stress-tolerant strains by chlorination might lead to more antimicrobial resistance [9].

## 3. Fundamentals of Chlorine Disinfection

Public water supply chlorination is one of the most significant advancements of the last century. Chlorine-based disinfectants have proven highly effective, safe, economical and widely accepted. Water hydrolysis, upon the addition of chorine, produces hypochlorous acid (HOCl), which dissociates yielding hydrogen and hypochlorite ions (Equations (1) and (2)).

Chlorine disinfection chemistry:(1)$${Cl}_{2}+H_{2}O\;⇌HOCl+H^{+}+{\;Cl}^{-}$$
(2)$$HOCl\;⇌{\;H}^{+}+{\;OCl}^{-}$$

Chlorine first reacts with organic and inorganic matters before pathogens inactivation. The amount of chlorine consumed in this process is termed chlorine demand. The combined chlorine that forms together with any free available chlorine in the water is called the chlorine residual (Figure 1). This is the component of the added chlorine that disinfects the water. Free available chlorine is formed by differences in the concentrations of hypochlorous and hypochlorite ions, a process that depends on the pH of the water [20]. Even though the chlorination procedure is well-researched, establishing an appropriate chlorine dose remains a difficult task for many field applications. Nevertheless, the effective chlorine dose should be sufficient to destroy pathogens and oxidize the organic contaminants as well as maintain sufficient free available chlorine in the water distribution system, post-chlorination [21].

### 3.1. Breakpoint Chlorination and Factors Influencing Disinfection

Breakpoint chlorination is the continuous addition of chlorine until the chlorine demand is satisfied, combined chlorine compounds are oxidized and only free chlorine remains. The four phases involved in breakpoint chlorination is depicted in Figure 2.

Complete oxidation of chlorine occurs in the first phase due to low chlorine concentration and lack of free chlorine residual. The phase is also known as the initial chlorine demand. Combined chlorine residual is formed between Cl^−^ and NH_4_^+^ in the second phase. The concentration of the added chlorine in this phase is proportional to the total concentration of chlorine residual. Continuous addition of chlorine in the third phase causes oxidation of the combined residues causing a reduced concentration of the residual chlorine up to “dip” or “breakpoint”. Beyond this point, the concentration of the residual chlorine increases [22]. 

Temperature, pH, contact time, density and the inherent nature of microorganisms present determine the efficacy of the chlorination. Keeping all other factors constant, higher microbial load will demand higher chlorine dose and contact time. The physiological state of the microbes plays an important role in their chlorine sensitivity [13]. Generally, autochthonous microbes in natural water environments are more resistant to chlorination than laboratory-grown strains.

### 3.2. Mechanisms of Chlorine Disinfection

The biocidal mechanism of chlorination remains poorly understood. Some investigators opined that HOCl and OCl^−^ formed when chlorine is added to water destroy microbial macromolecules. Further work on this supposition led to the “multiple hit” concept, which asserts that bacterial kill by chlorination is probably due to the attack and damage of microbial biomolecules [23]. Venkobachar et al. [24] and Haas [25] reported leakage of cellular macromolecules when bacteria were treated with chlorine. Venkobachar [24] also observed that chlorine significantly inhibits oxidative phosphorylation and the uptake of oxygen, and that, the inhibition of respiratory enzyme was responsible for the phosphorylation inhibition phenomenon, rather than a deficiency in phosphate uptake. In a different study, Chang [26] suggested that the extensive destruction of bacterial enzymatic systems is responsible for the rapid destruction of pathogens during chlorination. Various other investigators have also recognized that chlorine destroys microorganisms through changes in membrane permeability, nucleic acid damage and leakage of intracellular biomolecules [27,28]. Generally, it appears that chlorination causes physiological and morphological changes to the bacterial cell wall, altering its permeability. The chlorine molecules subsequently enter the cytoplasm, interfering with different enzymatic reactions, in a possible event cascade depicted in Figure 3.

### 3.3. Incidences of Chlorine Tolerant Microorganisms from Treated Water Sources

Various international and local authorities have regulations to ensure the protection of water sources from chemical and microbiological contaminants. At a global level, for instance, the WHO recommends chlorine residual of 0.2 to 0.5 mg/L in water supplies [29], while the South African Department of Water and Sanitation stipulates chlorine residual of 0.25 mg/L for discharged wastewater effluents released into surface water environment [30,31]. Although some concerns have emerged in recent years about the limitations of chlorine usage in water treatments, there is no doubt, however, that chlorination remains one of the most significant advances in water purification and public health protection [32]. Reports on chlorine surviving microorganisms from chlorinated water sources continue to emanate [33,34,35]. Various investigators have proposed different mechanisms by which microorganisms may develop tolerance to disinfectants. These include (i) cell surface modifications or encapsulation [36], (ii) attachment to surfaces or suspended particulate matter [37], (iii) microbial aggregation in biofilms [38], (iv) expression of multidrug efflux pump activity [9] and (v) spore formation [39]. These mechanisms conceivably favor the survival of opportunistic waterborne pathogens, thereby contributing to their persistence in water [40]. Table 1 lists some published studies on chlorine-resistant microorganisms from different sources and their reported mechanism of resistance.

## 4. Major Drivers of Antimicrobial Resistance (AMR) in Aquatic Environments

The prevalence of resistance genes in the aquatic environments is the result of a complex interplay of numerous factors propelling evolution by natural selections and variations, through cascades of mutational events. The buildup of chemical pollutants including antimicrobials, disinfectants, heavy metals, detergents, pharmaceuticals and xenobiotics contribute to the evolution and spread of resistance in the aquatic environment. Huge selective pressure resulting from anthropogenic activities escalates and enriches resistance determinants in microbial populations [51,52]. Efforts to reduce resistance development would include limiting the amount of resistant microbial loads as well as optimization of disinfection protocols in water treatment. It has been argued that AMR action plans may not achieve their anticipated objectives if all AMR drivers and pathways in the environment are not appropriately appraised.

### 4.1. Stepwise Accumulation of Drug Resistance Mutations

During antimicrobial administration, the application of doses within an MSW can allow the enrichment of a mutant fraction of the microbial population resulting in increased genetic and phenotypic resistance [53]. Genetic resistance is generally classified as acquired or de novo. Acquired resistance often occurs in a single step through the incorporation of resistance determinants frequently borne on mobile genetic elements from an external source. De novo resistance, on the other hand, arises either as a single step or as an accumulation of a series of mutations that discretely reduce susceptibility by a modest increment in a step-by-step manner (e.g., intermediate susceptibility). Usually, the next mutational step is more readily achieved with the occurrence of a previous mutation in a strain, since numerous alleles can accumulate [54]. In the stepwise process, a microbial cell is only considered resistant due to the presence of more than one genetic mutation [55]. This typical example is seen in *Streptococcus pneumonia* in the development of resistance against fluoroquinolones, where at least seven different alleles are identified [56].

### 4.2. Contribution of Chlorination to AMR Expansion via Horizontal Gene Transfer

Although chlorination is a vital step in water treatment, its contribution to the expansion and dissemination of AMR has been noted. Increased natural transformation rates were observed across different bacterial genera through the transfer of antimicrobial resistance genes (ARGs) and other mobile genetic elements (MGEs), leading to the enrichment of ARG after chlorine treatment [13]. The study also noted that the release of naked DNA (including plasmids, ARGs and integrons) from “killed” donor bacteria during the chlorination process contributed to an increased natural transformation of chlorine-injured-but-culturable strains at a rate >550 folds in comparison to untreated bacteria. In a related study, Guo and colleagues [57] reported that low chlorine doses ranging between 5 and 40 mg Cl min/L enhanced the rate of conjugative transfer of ARGs by factors between 2 and 5 due to increased induction of pilus on the surface of conjugative bacterial cells, facilitating the exchange of genetic materials. They, however, noted that high chlorine doses (>80 mg Cl min/L) significantly suppressed the rate of ARG transfer. In another study, a retrieval of chlorination-driven enrichment of ARGs between 18.1 and 102% was observed through metagenomics binning strategies [58]. In general, horizontal gene transfer plays a significant role in AMR expansion during water chlorination. 

## 5. Association between AMR and Disinfectant Resistance in Microorganisms

Putative resistance factors are thought to have existed in the natural environment, primarily for other functions apart from resistance conferment. However, the versatility of microbial pathogens has benefitted by recruiting and incorporating those genes into their own genome to evade the biocide effect of antimicrobials [59]. It should be noted that AMR genes are concurrently found with other genes that promote resistance to varieties of potentially harmful chemicals, including disinfectants in bacterial strains. The unfortunate historical convention, however, is the labelling of all these AMR genes; whereas, in actual fact, they can potentially promote resistance against different classes of chemicals, a phenomenon known as co-selection [60]. Co-selection of resistance has serious health implications and occurs either as co-resistance or cross-resistance. Cross-resistance allows a single resistance gene to foster protection against multiple biocides while co-resistance allows one gene to promote the maintenance of another resistance gene. Co-resistance phenomenon is exemplified as a “toolbox” in which one might only need one or two tools; however, the availability of many tools in the same box makes it easier to select any other tool for use should the need arise. Genomic architecture such as plasmids and transposons are the toolbox while the tools are the resistance genes co-existing on them [61]. The survival of some waterborne pathogens in chlorinated water and the co-selection for AMR raises concerns. Although there are conflicting reports on the role of chlorination in promoting AMR [9,62], it is noteworthy that a huge evidence-based gap still exists in substantiating the precise links between chlorination and antibiotic resistance in waterborne pathogens.

## 6. Proteome Mediated Chlorine Tolerance

During water treatment, the interactions of reactive chlorine with microbial biomolecules trigger several adaptive responses, which serve to prevent severe cellular damage, promoting resistance. Some of these include the release of detoxifying enzymes, chaperones activation, protein and DNA repair mechanisms, and changes in membrane conformation, among others. The activation of chaperone (heat-shock proteins, e.g., Hsp33, CnoX and RidA) for instance, prevents the misfolding of other proteins and the formation of lethal protein aggregation in response to the accumulation of protein unfolding intermediates caused by chlorine disinfection [63]. Hsp33 is the most studied chaperone, which is specifically activated as a response to the production of unfolding proteins by chlorination, although initiated to a lesser degree by other oxidizing agents [64,65]. Similarly, a number of transcriptional regulators, including HypT, NemR and RclR, which are crucial for microbial survival, are activated by oxidizing agents [66]. However, the specific roles of the genes regulated by these factors remain unclear. Overall, chlorine resistance is believed to be mediated by oxidative stress regulons, universal stress response rpoS-regulon and heat shock proteins. Wang and colleagues [67] noted that there is overlap between the putative mechanisms of heat resistance and chlorine resistance in *E. coli*, although the extent to which this overlap occurs has not been confirmed experimentally.

## 7. Biofilm and Pathogen Survival in Water Treatment

Biofilm is an adaptive mechanism of disinfection tolerance and has a long history in water treatment. The cells in a biofilm constantly interact and coordinate their activities efficiently by cell-to-cell contact or indirectly, via the release of signaling molecules, a process known as quorum sensing (QS) [68]. The biofilm microbial consortia are shielded by the extracellular polymeric substances (EPSs), thereby protecting them from chlorine, antimicrobials, increased temperatures and other stressful conditions. Upon maturation, portions of the biofilm can detach and be delivered to the end-users, which may permit a possible outbreak of waterborne disease [69]. Unfortunately, it is almost impossible to totally eradicate biofilm formation in water treatment and distribution systems; therefore, a number of factors to control their growth should be carefully considered. Some of these factors include nutrient availability in water, surface topography and pipe materials, hydrodynamics of the distribution network, pH, temperature, and disinfectant residual [70]. Figure 4 illustrates the various stages involved in the biofilm microbiological buildup.

### Resistance and Pathogen Protection in Biofilms

Microbial biofilm resistance has numerous health, environmental and economic implications, such as water distribution system clogging, biofouling, food-processing disruptions, oil recovery, decay of water pipes and clinical implants, among others [71]. Microbial cells embedded in biofilms often show varying phenotypic characteristics, such as increased resistance to biocidal agents compared to planktonic cells. The exact mechanism of biofilm resistance remains unclear. The process, however, has been described as multifactorial involving the spatial distribution and organization of the biofilm architecture [38]. While numerous studies have concentrated on biofilm resistance to antibiotics [72,73,74], very minimal information exists on biofilm resistance to other biocides including disinfectants such as chlorine. Generally, resistance to biocides may be phenotypic (tolerance), genotypic or intrinsic in nature. For instance, the multilayered EPSs and microbial cells in a biofilm may constitute a very complex and compact barrier, making it difficult for the biocide to penetrate and reach the inner cells, thereby hampering its efficacy [75]. Additionally, glycocalyx or capsule, found in both Gram-positive and Gram-negative bacteria, can accumulate biocide molecules up to 25% of its weight [74], and can provide protection against antimicrobials in a biofilm [73]. 

Apart from the physical and structural barriers which biofilm provides, the preferential expression of some specific genes within biofilms can contribute to resistance. The upregulation of the *ndvB* gene that codes for a glucosyltransferase, an enzyme that catalyzes the production of periplasmic β-(1→3)-cyclic glucans, for example, is specifically noted in biofilms [76]. The glucans presumably sequester antibiotics away from their cellular target in the periplasm [77]. Moreover, *ndvB* upregulation is involved in the expression of some biofilm-specific genes associated with ethanol oxidation, and these genes may additionally play specific roles in biofilm resistance. Even though the upregulation of the *ndvB* gene is associated with biofilm, the mechanisms involved in its control and regulations are not known [78]. Understanding the genetic and physiological requirements for this biofilm-specific gene expression may serve as a potential tool in reducing biocide and antimicrobial resistance.

## 8. Multidrug Efflux Pumps Induction and Biocide Resistance

Some efflux pumps are expressed constitutively at low levels, thereby contributing to intrinsic resistance in microbial cells. Active efflux pumps may also contribute to high levels of resistance due to their overexpression in the presence of an effector molecule. Overexpression of multidrug efflux pumps may, however, be transient in nature [79]. While efflux pumps have mostly been associated with antibiotic resistance in pathogens, it is worthy to note that they can non-selectively extrude diverse compounds, including heavy metals, antiseptics, disinfectants, dyes, toxins, organic pollutants, metabolites, quorum sensing signaling molecules, neurotransmitters and quaternary ammonium compounds [80,81]. Structurally, an efflux pump is made up of four major components of an outer membrane protein, a middle periplasmic protein, an inner membrane protein and a transmembrane duct. As depicted in Figure 5, the outer and the inner membranes stabilize the channel (duct) in a closed state through their interaction with the periplasmic membrane protein [82]. The system is activated when a substance is transported, e.g., drug binds to the inner membrane protein, which then activates a cascade of biochemical events leading to the opening of the channel and eventual expulsion of the drug molecules. The protein-protein interaction generates the required energy by exchanging the molecule for a hydrogen ion (H^+^) [83]. Depending on the secondary structure, energy source, homology of amino acids and size, active efflux systems are classified into five different families (Figure 5). These include the ATP-binding cassette (ABC), the major facilitator superfamily (MFS), the resistance-nodulation division superfamily (RND), the small multidrug resistance family (SMR) and the multi antimicrobial and toxic compound extrusion protein family (MATE) [84]. 

The MFS transporters are also referred to as uniporter-symporter-antiporter (USA) family and represent the largest efflux family with an extraordinary broad-spectrum substrate affinity [85]. The SMR proteins are one of the smallest efflux systems with lengths ranging from about 100 to 140 amino acids. They are often inherited chromosomally but have also been found encoded by prophages, conjugative plasmids as well as Class I and Class III integrons. *E. coli* ethidium multidrug resistance protein E (EmrE) is one of the most characterized members of this protein family [80]. The most intensively characterized RND protein efflux system is the AcrAB-TolC, described in Gram-negative bacteria, including *E. coli* K-12 and *Salmonella enterica* serovar Typhimurium SH5014. It is made up of a periplasmic protein, AcrA, anchored to the inner membrane, twelve transmembrane α-helices and two large hydrophilic loops AcrB transporters, and the TolC outer membrane protein, which forms a channel allowing substrates to diffuse across the outer membrane. Most members of the RND family contain large polypeptides with 700–1300 amino acid residues [86]. The MATE proteins were initially identified in *Vibrio parahaemolyticus* and, subsequently, in many other cells, including humans [87]. MATE majorly transports cationic substrates, including clinically important drugs. The ABC transporters are energy-dependent and are found in all life forms. Seven distinct subfamilies of these transporters have been classified phylogenetically from ABCA to ABCG [88]. The proteins can transport a wide range of substrates and are characterized by two nucleotide-binding domains and two transmembrane domains. Overexpression of the ABC transporter proteins has been associated with increased extrusion of chemotherapeutic agents and multidrug resistance cases [89].

The expression of efflux is tightly regulated, and its induction can be achieved either in the presence of a right inducer molecule transiently or by mutations of the genes which downregulate their expression constitutively [90]. While a single efflux system is capable of conferring resistance to multiple biocidal agents, simultaneous induction of multiple efflux pumps has been identified in bacteria strains such as *P. aeruginosa* and *Stenotrophomonas maltophilia* strains, thus conferring both unique and overlapping biocide and drug selectivity [91,92]. Efflux systems are a growing clinical concern due to the possible co-existence of an assortment of diverse pump systems in clinically relevant microbial pathogens. The systems can also be easily spread on conjugative plasmids and other mobile genetic elements further underlining their importance in resistance biology. The redundant (overlapping) substrate recognition and drug polyspecificity nature of the efflux pump systems, coupled with their cross-resistance activities to varieties of biocides and antimicrobials are particularly prominent in Gram-negative pathogens such as the Enterobacteriaceae family. These group of enteric bacteria can survive and persist in soil, and aquatic environments, and are able to acquire biocide adaptation through chronic sub-lethal concentration exposure [80]. Due to the important role active efflux mechanisms play in resistance, the development of antimicrobials that target efflux systems and effectively inhibit their activities has been considered a potential control strategy to tackle the problem of resistance in microorganisms.

## 9. Conclusions and Future Directives

The endless problem of resistance proliferation among clinically relevant waterborne pathogens is a worrisome topic that necessitates the improvement of contemporary and next-generation treatment procedures. Since the turn of the century, large-scale chlorination has helped in maintaining the microbiological quality of public water supplies. Even though this has recorded many gains, conflicting reports continue to emanate on the role chlorination plays in the enhancement of AMR genes and resistant microorganisms in aquatic environments. While chlorine has been effective in reducing outbreaks of waterborne diseases, the molecular mechanisms underlining its roles in promoting the evolution and spread of resistance remain a grey area, with its attendant ecological and public health implications. Cross-resistance and co-resistance are important phenomena in the associations between chlorine and AMR. Similarly, proteome-mediated chlorine resistance has been described in some bacterial species while non-specific expression of multidrug efflux mechanisms and biofilm formation are other processes identified in waterborne pathogen resistance. Despite the various efforts toward understanding disinfection resistance among waterborne pathogens, a huge evidence-based gap at substantiating the exact links between chlorine disinfection and AMR spread exists. Unfortunately, there is still no standardized method to evaluate microbial chlorine resistance making the comparison of chlorine resistance challenging. In light of the above and the widespread observation of reduced disinfection efficacy of chlorine at the recommended dose, further investigation is crucial to more accurately explicate chlorine resistance mechanisms and the relationship between AR and chlorine resistance/tolerance for more rational water disinfection approaches with a view to improve its efficacy or develop novel strategies.

## Figures and Tables

**Figure 1 antibiotics-11-00564-f001:**
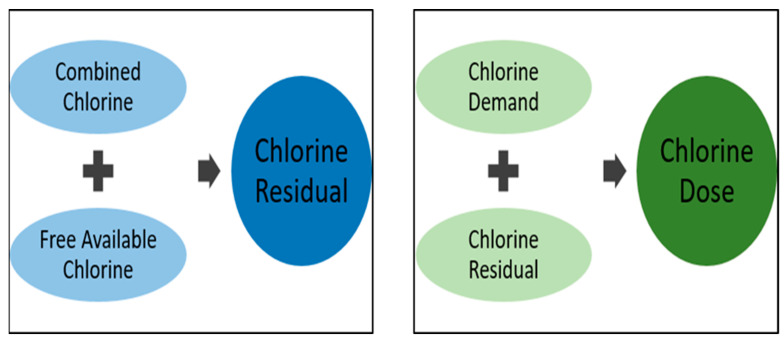
Schematic illustration of chlorine disinfection.

**Figure 2 antibiotics-11-00564-f002:**
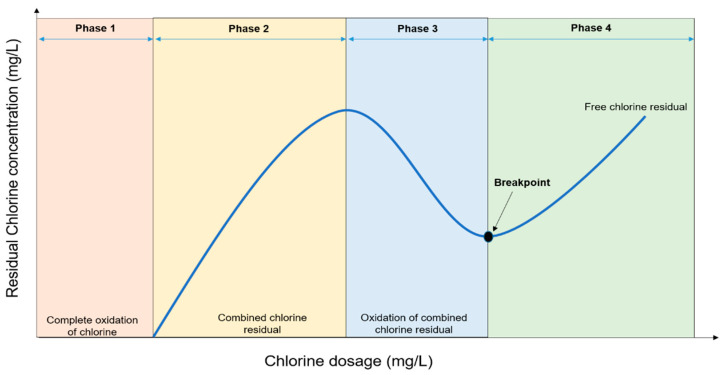
Graphical representation of breakpoint chlorination.

**Figure 3 antibiotics-11-00564-f003:**
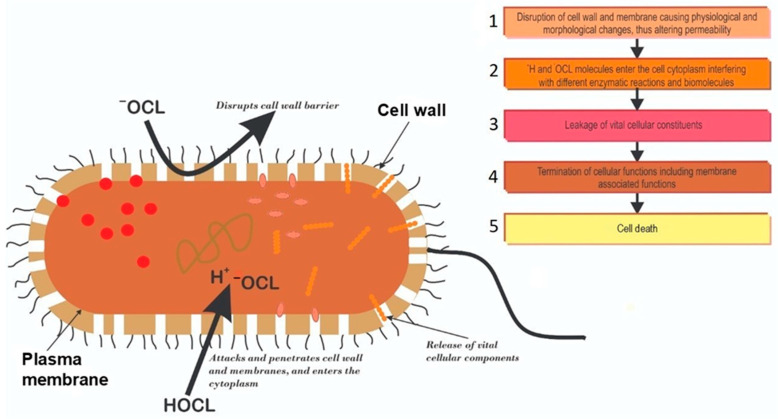
Schematic presentation of the possible events cascade of chlorine disinfection leading to microbial cell death. 1: Dissociating chlorine molecules attack and destroy bacterial cell walls altering permeability; 2: Entry of chlorine molecules into the cytoplasm interfering with biomolecules and enzymatic reactions; 3: Altered permeability causes vital cellular components leakage; 4: The series of events lead to the loss of cellular constituents and functions; 5: The loss of cellular constituents and function lead to eventual cell death.

**Figure 4 antibiotics-11-00564-f004:**
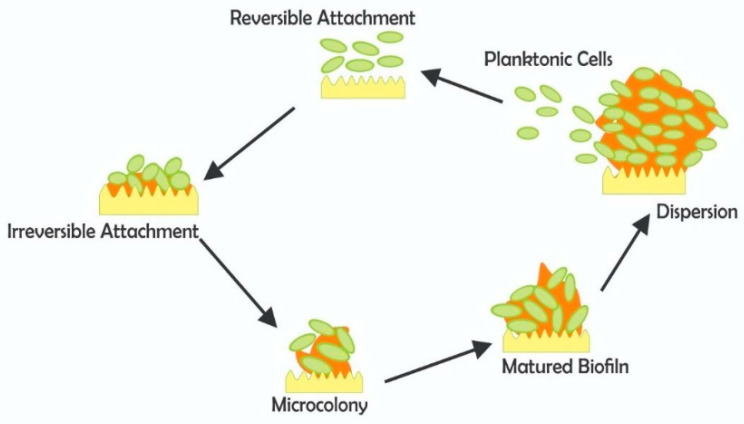
Schematic illustration of different phases in biofilm microbiology.

**Figure 5 antibiotics-11-00564-f005:**
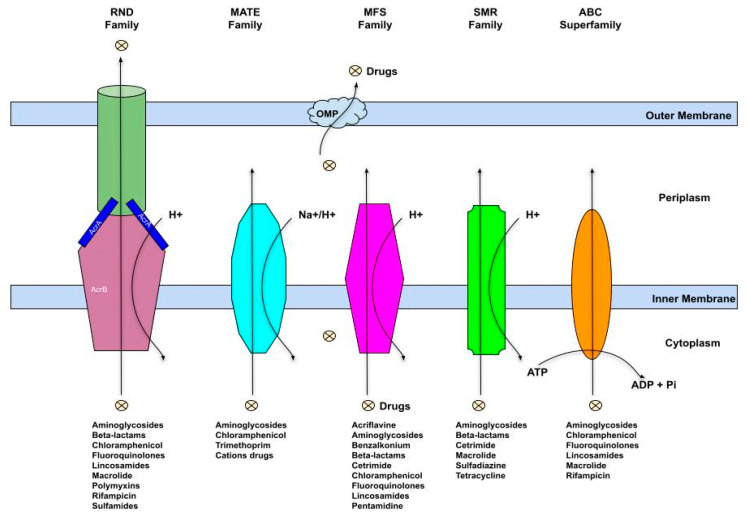
Different classes of efflux systems.

**Table 1 antibiotics-11-00564-t001:** Summary of some reports of disinfectant-resistant microbes isolated from diverse aquatic sources.

Source	Microorganism	Disinfectant Concentration/Time	Mechanism(s) of Resistance	Reference
Drinking water	*P. aeruginosa*	≤0.5 mg/L Cl^−^	Natural resistance due to the permeability barrier caused by outer membrane lipopolysaccharides; biofilm formation	[41]
Experimental isolates	*Acinetobacter* *baumannii*	0.2–4 mg/L	Increased expression of efflux pumps other antibiotic resistance genes	[42]
Drinking water reservoir	*Acinetobater* species, *Serratia* species	2 mg/L	Not determined	[35]
Sewage	*Bacillus* species	0.1 mg/L NaOCl	Probable spore formation	[43]
Secondary effluent	*Citrobacter* species	0.5 mg/L Ca(OCl)_2_ for 30 min	Not determined	[44]
Drinking water	*Bacillus* species, Actinomycete	10 mg/L NaOCl for 2 min	Cellular aggregation or adhesion to suspended particulate. Production of extracellular slime or capsular material	[45]
Drinking water and experimental isolates	Heterotrophic bacteria, faecal coliforms, *E. coli*, *Salmonella typhimurium*, *Yersinia enterocolitica*, *Shigella sonnie*	2.0 mg/L free chlorine for 1 h	Bacterial attachment to surface and production of extracellular slime layer	[46]
Chlorine-demand–free buffer solution	Coliform isolated from drinking water systems and Enteric bacterial from culture collections cocultured with protozoa (Ciliates and amoebae).	2–4 mg/L free chlorine for 1–2 h	Shielding of bacteria from chlorine by ingesting protozoans (cysts) and, thus, enhancing resistance	[47]
Treated drinking water	*S. aureus*, *Micrococcus* varians, *Aeromonas hydrophila*	1–100 mg/L Ca(OCl)_2_ solution for 30 min	Possible synthesis of unique proteins or aggregation of bacteria or encapsulation	[33]
Environmental isolates (Wastewater clarifier effluent) suspended in phosphate buffer saline	*Enterococcus* species	0.5 mg/L Ca(OCl)_2_ for 30 min	Not determined	[48]
Environmental strains cultured in sterile phosphate buffer solution	*Legionella pneumophila* from environmental water cocultured with *Acanthamoeba* species	2–3 mg/L Cl2 for 1 h with a residual Cl2 of 1 mg/L after 1 h	Possible phenotypic modification of *Legionella pneumophila* due to intra-cellular growth with *Acanthamoeba* sp.	[49]
Environmental isolates from wastewater treatment plants suspended in saline	*Bacillus* species, *Citrobacter freundii*, *Enterobacter* species, *Kluyvera cryocrescens*, *Kluyvera intermedia*	0.5 mg/L NaOCl for 30 min	Authors suggested the possible expression of certain stress factor genes which may reduce bacterial metabolism or change the permeability of cell membranes	[50]
Environmental isolates (Wastewater clarifier effluent) suspended in phosphate buffer saline	*E. coli*	0.5 mg/L NaOCl for 30 min	Not determined	[34]

## Data Availability

Not applicable.

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
