# Peer review of "Does Chlorination Promote Antimicrobial Resistance in Waterborne Pathogens? Mechanistic Insight into Co-Resistance and Its Implication for Public Health"

_antibiotics, 2022, doi:10.3390/antibiotics11050564_

Round 1
Reviewer 1 Report
The ms "Does Chlorination Promote Antimicrobial Resistance in Water-borne Pathogens? Mechanistic Insight into Co-resistance and its Implication for Public Health" describes various factors and mechanisms underlying the emergence and spread of resistance and the possible association with chlorination.
The ms is divided into well-organised sections, describing several factors and mechanisms possibly associated with chlorination and AMR.
The ms is overall well written with a few typos (lines: 44,66, 131, 148, 202, and 289(missing :)).
Regarding the figures, I would suggest changing figure 3. It would be clearer if the figure was all visual. Meaning: a cell schematic for all the text boxes included in the figure. Or just add a number in the process and describe it in the figure legend.
Author Response
Response to Reviewer 1 Comments
Point 1: The ms is overall well written with a few typos (lines: 44,66, 131, 148, 202, and 289(missing :)).
Response1: The typos identified by the reviewer have been revised as follows: line 44: ‘antimicrobial drug-microbe’ has been revised as ‘antimicrobial-microorganisms’.
- line 66: The statement has been revised as “A comprehensive understanding of the molecular basis of resistance development in bacteria will significantly contribute to these agents' more rational selection and utilization toward curbing AMR proliferation and advancing the global antimicrobial stewardship effort”.
- line 131: The typo has been corrected. The ‘Break point’ preceding the sentence has been deleted.
- line 202: The sentence has been revised as: The unfortunate historical convention, however, is the labelling of all these AMR genes, whereas, in actual fact, they can potentially promote resistance against different classes of chemicals, a phenomenon known as co-selection [58].
- line 289: The typo has been corrected and the sentence revised as follows: While efflux pumps have mostly been associated with antibiotic resistance in pathogens, it is worthy to note that they can non-selectively extrude diverse compounds, including heavy metals, antiseptics, disinfectants, dyes, toxins, organic pollutants, metabolites, quorum sensing signalling molecules, neurotransmitters and quaternary ammonium compounds [78,79].
Point 2: Regarding the figures, I would suggest changing figure 3. It would be clearer if the figure was all visual. Meaning: a cell schematic for all the text boxes included in the figure. Or just add a number in the process and describe it in the figure legend.
Response 2: As suggested by the reviewer, Figure 3 has been revised. Numbers have been added, and the process is described in the figure legend.

Reviewer 2 Report
Chlorination of public water supplies has been one of the most important advances in drinking water safety in the last century. Chlorine-based disinfectants have been shown to be efficient, safe, economical, and widely accepted. Long-term use or inappropriate use of chlorine to disinfect water leads to the production of drug-resistant pathogens, which endanger public safety. It is a serious problem of health and safety in the world. The authors focused on the concept of the mutational selection window (MSW), within which doses applied during antimicrobial administration can enrich the mutant portion of the microbial population, leading to increased genetic and phenotypic resistance. This is generally believed to be the cause of drug resistance. In the process of causing drug-resistance, there are results of current research, such as Mechanisms of Chlorine Disinfection, Incidences of Chlorine Tolerant Microbes from treated Water Sources, Major Drivers of Antimicrobial Resistance (AMR) in Aquatic Environments, Association between AMR and Disinfectant Resistance in Microorganisms, Proteome Mediated Chlorine tolerance, Biofilm and Pathogen Survival in Water Treatment, Multidrug Efflux Pumps Induction and Biocide Resistance, etc., have a clear overall understanding and concept of this aspect. The current achievements and practical gaps are clarified, and future researchers are encouraged to study or discuss a feasible implementation plan based on the existing information, such as standard methods for detecting drug-resistant pathogenic bacteria, new drug targets, appropriate usage and usage time of the recommended amount of chlorine for disinfection, etc.
The topic is relevant in the field of chlorination of public water. Most of the research on bacterial resistance comes from the study of antibiotics, and there are few reports on the resistance problem caused by chlorine disinfection. The importance of this review is based on information from antibiotic resistance research combined with information on resistance due to chlorine disinfection. It is hoped that the integration of this information will provide subsequent researchers with more perspectives and a better basis for future problem solving.
3.2. Mechanisms of Chlorine Disinfection
Breakpoint The biocidal mechanism of chlorination remains poorly understood. Please rewrite the sentence.
3.3. Incidences of Chlorine Tolerant Microbes from treated Water Sources s
s should be deleted
Author Response
Response to Reviewer 2 Comments
Point 1: 3.2. Mechanisms of Chlorine Disinfection
Breakpoint The biocidal mechanism of chlorination remains poorly understood. Please rewrite the sentence.
Response 1: The sentence has been revised accordingly. The ‘Break point’ preceding the sentence has been deleted.
Point 2: 3.3. Incidences of Chlorine Tolerant Microbes from treated Water Sources s
s should be deleted
Response 2: the ‘s’ has been deleted.

Reviewer 3 Report
I have recently had the opportunity to revise several review paper and unfortunately the conclusion was usually not positive, as the manuscripts did not contribute much to the current state of the art. This time is completely the opposite and it was a real pleasure to read the following manuscript. Almost 42% cited papers was published in in the last 5 years. The manuscript is generally well organized and comprehensively described. The most important discussed aspects are illustrated with eye-catching figures. Additionally, table is included to make the study of the incorporated content more convenient. My recommendation is that this manuscript should be accepted for publication but some points should be improved:
Line 162. In my opinion spore formation cannot be acknowledged as developed mechanisms of tolerance to disinfectants. Spore formation is specific property of some bacteria such as the Bacillus and Clostridium species. These bacteria are highly resistant to physical and chemical influences and of course by forming spores bacteria can survive adverse environmental conditions by “reawakening” when favourable conditions return. Escherichia coli are non-spore forming bacteria and cannot create this mechanism of resistance.
Subsection 4.1. Authors should depicted a relationship between drug resistance mutations and chlorination. There seems to be that in this section other mechanism of horizontal gene transfer (HGT) such as conjugation, transformation and transfection should be also described more precisely with reference to chlorination. Some aspects are presented in ref. [13] Jin, M., Liu, L., Wang, Dn. et al. Chlorine disinfection promotes the exchange of antibiotic resistance genes across bacterial genera by natural transformation. ISME J 14, 1847–1856 (2020) and Guo MT, Yuan QB, Yang J. Distinguishing effects of ultraviolet exposure and chlorination on the horizontal transfer of antibiotic resistance genes in municipal wastewater. Environ Sci Technol. 2015 May 5;49(9):5771-8 and Liping Ma, Huiying Yang, Lei Guan, Xiaoyu Liu, Tong Zhang,Risks of antibiotic resistance genes and antimicrobial resistance under chlorination disinfection with public health concerns, Environment International, 158, 2022, 106978.
In table 1 I suggest use the same units for disinfectant concentration e.g. mg/L
verse 10 – should be Ca(OCl)2 (subscript)
verse 11 – should be Cl2 (subscript) and Legionella pneumophila and Acanthamoeba sp. written in italics
But the funniest is that in conclusion section, the authors did not answer the question posed in the title: Does chlorination promote antimicrobial resistance in waterborne pathogens?
Author Response
Response to Reviewer 3 Comments
Point 1: Line 162. In my opinion, spore formation cannot be acknowledged as a developed mechanism of tolerance to disinfectants. Spore formation is a specific property of some bacteria, such as the Bacillus and Clostridium species. These bacteria are highly resistant to physical and chemical influences and of course by forming spores bacteria can survive adverse environmental conditions by “reawakening” when favourable conditions return. Escherichia coli are non-spore-forming bacteria and cannot create this mechanism of resistance.
Response 1: We appreciate the observation of the reviewer and had this opinion in our thought. However, while it is well acknowledged that E. coli is non-spore-forming and used as a surrogate for faecal contamination; its presence in water is treated as an increased probability of the presence of other pathogens, including other enterics and possibly spore-forming bacteria. It should be noted that the mechanisms discussed in this section are not specific to indicator bacteria such as E. coli, but also to other pathogens that are encountered in contaminated water generally and mostly those that have been identified in association with biofilm and other resistance structures. In addition to this, and depending on their inherent nature or intrinsic capabilities, different strains may adopt different tolerance/survival strategies to circumvent environmental stress including disinfection.
Point 2: Subsection 4.1. Authors should depicted a relationship between drug resistance mutations and chlorination. There seems to be that in this section other mechanism of horizontal gene transfer (HGT) such as conjugation, transformation and transfection should be also described more precisely with reference to chlorination. Some aspects are presented in ref. [13] Jin, M., Liu, L., Wang, Dn. et al. Chlorine disinfection promotes the exchange of antibiotic resistance genes across bacterial genera by natural transformation. ISME J 14, 1847–1856 (2020) and Guo MT, Yuan QB, Yang J. Distinguishing effects of ultraviolet exposure and chlorination on the horizontal transfer of antibiotic resistance genes in municipal wastewater. Environ Sci Technol. 2015 May 5;49(9):5771-8 and Liping Ma, Huiying Yang, Lei Guan, Xiaoyu Liu, Tong Zhang,Risks of antibiotic resistance genes and antimicrobial resistance under chlorination disinfection with public health concerns, Environment International, 158, 2022, 106978.
Response 2: Response 2: As suggested by the reviewer, a new Subsection 4.2 has been included in the manuscript to depict the relationship between the role of horizontal gene transfer and bacterial antimicrobial resistance expansion. The suggested references also have been carefully reviewed and used in updating the arguments presented
Point 3: In table 1 I suggest using the same units for disinfectant concentration e.g. mg/L
Response 3: All the units have been harmonized and reported as mg/L as suggested by the reviewer.
Point 4: verse 10 – should be Ca(OCl)2 (subscript)
Response 4: The compound name has been revised as recommended by the reviewer.
Point 5: verse 11 – should be Cl2 (subscript) and Legionella pneumophila and Acanthamoeba sp. written in italics
Response 5: The compound name has been revised as recommended by the reviewer, and the nomenclatures italicized.
Point 6: But the funniest is that in the conclusion section, the authors did not answer the question posed in the title: Does chlorination promote antimicrobial resistance in waterborne pathogens?
Response 6: We appreciate the opinion of our reviewer and understand his/her perception. It is should be noted that based on the state of the art and the most recent information on the subject of discourse, there is increasing evidence, as reported by some of the literature cited in this review that cases of chlorine tolerance/resistance are being documented which in the long run have implication for antimicrobial resistance in waterborne microbial pathogens. At the same time, we should take into cognizance the view/opinions of other authors who have presented contrary arguments to this observation. Ultimately, this lack of consensus on the exact role of chlorination in antimicrobial resistance promotion/enhancement remains an important knowledge gap as identified in this paper and undoubtedly requires further careful investigations. Notwithstanding, a number of co-resistance/cross-resistance mechanisms have been suggested/identified by various authors which are associated with chlorine tolerance/resistance and antimicrobial resistance expansion.
